# The Management of Peutz–Jeghers Syndrome: European Hereditary Tumour Group (EHTG) Guideline [note 1]

**DOI:** 10.3390/jcm10030473

**Published:** 2021-01-27

**Authors:** Anja Wagner, Stefan Aretz, Annika Auranen, Marco J. Bruno, Giulia M. Cavestro, Emma J. Crosbie, Anne Goverde, Anne Marie Jelsig, Andrew R. Latchford, Monique E. van Leerdam, Anna H. Lepisto, Marta Puzzono, Ingrid Winship, Veronica Zuber, Gabriela Möslein

**Affiliations:** 1Department of Clinical Genetics, Erasmus MC Cancer Institute, University Medical Center Rotterdam, 3000CA Rotterdam, The Netherlands; a.goverde.1@erasmusmc.nl; 2Institute of Human Genetics, Medical Faculty, University of Bonn, 53127 Bonn, Germany; stefan.aretz@uni-bonn.de; 3National Center for Hereditary Tumor Syndromes, University Hospital Bonn, 53127 Bonn, Germany; 4Department of Obstetrics and Gynecology and Tays Cancer Center, Tampere University Hospital, 33520 Tampere, Finland; anaura@utu.fi; 5Department of Gastroenterology & Hepatology, Erasmus MC Cancer Institute, University Medical Center Rotterdam, 3000CA Rotterdam, The Netherlands; m.bruno@erasmusmc.nl; 6Division of Experimental Oncology, Gastroenterology and Gastrointestinal Endoscopy Unit, Vita-Salute San Raffaele University, IRCCS San Raffaele Scientific Institute, 20132 Milan, Italy; cavestro.giuliamartina@hsr.it (G.M.C.); Puzzono.Marta@hsr.it (M.P.); 7Department of Gynecology, Manchester University NHS Foundation Trust, Manchester Academic Health Science Centre, Manchester M13 9WL, UK; Emma.Crosbie@manchester.ac.uk; 8Division of Cancer Sciences, Faculty of Biology, Medicine and Health, University of Manchester, St Mary’s Hospital, Manchester M13 9WL, UK; 9Department of Clinical Genetics, University Hospital of Copenhagen, 2100 Copenhagen, Denmark; anne.marie.jelsig@regionh.dk; 10Department of Surgery and Cancer, Imperial College London, London SW7 2AZ, UK; andrew.latchford@nhs.net; 11Polyposis Registry, St. Marks Hospital, London HA1 3UJ, UK; 12Department of Gastro-intestinal Oncology, Netherlands Cancer Institute, 1006BE Amsterdam, The Netherlands; m.v.leerdam@nki.nl; 13Department of Gastroenterology and Hepatology, Leiden University Medical Center, 2300RC Leiden, The Netherlands; 14Department of Surgery, University Hospital of Helsinki, 00029 Helsinki, Finland; anna.lepisto@hus.fi; 15Department of Genomic Medicine, The Royal Melbourne Hospital, University of Melbourne, Melbourne 3052, Australia; Ingrid.Winship@mh.org.au; 16Breast Surgery Unit, IRCCS San Raffaele Scientific Institute, 20132 Milan, Italy; zuber.veronica@hsr.it; 17Center for Hereditary Tumors, Ev. BETHESDA Khs. Duisburg, Academic Hospital University of Düsseldorf, 47053 Duisburg, Germany; gmoeslein@outlook.de

**Keywords:** Peutz–Jeghers syndrome, *STK11*, guideline

## Abstract

The scientific data to guide the management of Peutz–Jeghers syndrome (PJS) are sparse. The available evidence has been reviewed and discussed by diverse medical specialists in the field of PJS to update the previous guideline from 2010 and formulate a revised practical guideline for colleagues managing PJS patients. Methods: Literature searches were performed using MEDLINE, Embase, and Cochrane. Evidence levels and recommendation strengths were assessed using the Grading of Recommendations Assessment, Development and Evaluation (GRADE). A Delphi process was followed, with consensus being reached when ≥80% of the voting guideline committee members agreed. Recommendations and statements: The only recent guidelines available were for gastrointestinal and pancreatic management. These were reviewed and endorsed after confirming that no more recent relevant papers had been published. Literature searches were performed for additional questions and yielded a variable number of relevant papers depending on the subject addressed. Additional recommendations and statements were formulated. Conclusions: A decade on, the evidence base for recommendations remains poor, and collaborative studies are required to provide better data about this rare condition. Within these restrictions, multisystem, clinical management recommendations for PJS have been formulated.

## 1. Introduction

Peutz–Jeghers syndrome (PJS) is a rare hereditary condition characterized by mucocutaneous pigmentation and Peutz–Jeghers hamartomatous polyps, predominantly affecting the small intestine (Figure 1) [1,2]. The diagnostic clinical criteria for Peutz–Jeghers syndrome are shown in Table 1 [3,4,5]. In childhood, symptoms are mostly caused by polyp-related complications, including bleeding, anaemia, and obstructive symptoms. Small bowel intussusception is the most urgent and even life-threatening manifestation. In adulthood, PJS patients face an increased risk of a constellation of different cancers.

PJS is caused by heterozygous germline pathogenic variants (PV) in the serine threonine kinase 11 tumor suppressor gene (*STK11/LKB1* gene) and follows an autosomal dominant inheritance pattern [7,8]. Individuals suspected to have PJS should be offered genetic counselling and genetic testing, with informed consent or informed assent for children. Once a disease-causing variant is detected in an individual with PJS, at-risk relatives can be tested for this variant in the context of pre- and post-test genetic counselling. Tailored surveillance should be offered to all PV carriers in order to manage their risks of intussusception and malignancies.

In view of the multisystem nature of PJS, the care for PJS patients and their relatives requires multidisciplinary expertise. Unfortunately, there is a relative paucity of clinical and scientific data to guide PJS management. The available evidence was reviewed by a diverse group of medical specialists with complementary skills in the field of PJS in order to formulate a practical set of guidelines for colleagues managing patients with PJS.

## 2. Methods

The European Hereditary Tumour Group (EHTG) commissioned this guideline (chair GM) and appointed a guideline leader (AW), who invited the listed authors to participate in guideline development. Two to three members around each key topic formulated key questions that were approved by the other members (see Appendix A). To develop the guideline, the committee members had a live meeting, telephone conferences, and online discussions from July 2019 until November 2020. Searches were performed in MEDLINE, Embase, and Cochrane, and articles were selected through title and abstract screening followed by full-text screening (see Appendix A). Expert members presented the results of the search and proposed statements to all members of the guideline committee. Evidence levels and recommendation strengths were assessed using the Grading of Recommendations Assessment, Development and Evaluation (GRADE) [9]. Since literature on Peutz–Jeghers syndrome is limited, a Delphi procedure was organized within the guideline committee, which comprised two rounds to gain consensus [10]. All guideline committee members, except for MP, who assisted in the literature search, completed the online Delphi questionnaire. The level of agreement with statements was rated using a seven-point Likert scale: “Very strongly agree”, “Strongly agree”, “Agree”, “Neither agree nor disagree”, Disagree”, “Strongly disagree”, or “Very strongly disagree” [11]. If the statement was not their area of expertise, participants had the option to opt out. Participants were asked if the statements were clear and suggested improvements where they were not. After the Delphi rounds, the statements were discussed and adjusted if necessary, during online sessions. Consensus was reached when ≥80% of the voting guideline committee members had voted either “Very strongly agree”, “Strongly agree”, or “Agree” during the 2 rounds of Delphi.

This guideline was issued in 2020 and will be considered for (partial) updating if indicated. Updates will be noted on the EHTG website: http://www.ehtg.org/guidelines/.

## 3. Cancer Risks in Peutz–Jeghers Syndrome

In 2000, the study by Giardello et al. clearly indicated that PJS patients are at considerable risk of developing cancer [12]. This was confirmed by later studies describing overall lifetime cancer risks of 55–85% [13,14,15,16,17,18,19]. An overview of the published studies on cumulative cancer risks in PJS is shown in Table 2. Caution about these risks needs to be exercised; due to the generally small numbers of patients and very wide confidence intervals, these data are difficult to interpret, and the true risks are difficult to estimate. The described risks are likely to be an overestimation due to retrospective analysis and selection bias.

## 4. Recommendations and Statements

Based on the reviewed literature and expert opinion, recommendations and statements were formulated on (1) clinical genetic, (2) gastrointestinal, (3) surgical, (4) pancreatic, (5) breast, and (6) gynecological management.

### 4.1. Clinical Genetic Management

*STK11 (LKB1)* (MIM 602216) is the only gene that is known to be associated with PJS. It is located on 19p13.3 and was identified by linkage analysis followed by cloning of the gene and identification of heterozygous PVs in affected individuals [7,8,20,21]. The vast majority of PVs detected are truncating (loss of function) variants, including nonsense, splice site, smaller insertions and deletions, as well as deletions of one or multiple exons.

The detection rate of *STK11* PVs in cohorts of PJS patients varies. Without the use of methods to identify large deletions and duplications such as Multiplex Ligation dependent Probe Amplification (MLPA), detection rates of 10–70% were reported. With the introduction of MLPA, it became clear that larger deletions account for a significant proportion of PVs, and higher detection rates of 60–100% were reported (Appendix A) [13,22,23,24,25,26,27,28,29,30,31,32,33,34,35,36,37,38,39,40,41]. The clinical criteria used for the enrollment of patients in these studies vary; some used the clinical criteria by Tomlinson et al. and Beggs et al., while others refer to the World Health Organization (WHO) criteria (Table 1) [3,4,5].

Genetic heterogeneity has been suggested on the basis that PVs are not found in all patients with PJS; however, despite considerable effort, no other gene has been associated with PJS so far [22,31,42,43,44,45]. Patients without detectable PVs might be explained by variants in non-coding sequences, limitations in technique, misdiagnosis, as well as mosaicism—the latter having been reported in a few case reports [46,47]. Screening for *STK11* mosaicism in blood or other tissue could be considered in patients who fulfill clinical criteria for PJS but without a detectable PV by initial genetic screening.

The detection of a PV enables predictive genetic testing of at-risk relatives. Furthermore, reproductive diagnostic options such as preimplantation genetic diagnosis (PGD) become available once a PV in *STK11* is identified in a PJS patient.

Initial genetic screening in patients who do not meet the clinical criteria for PJS is recommended by the European Society for Pediatric Gastroenterology Hepatology and Nutrition (ESPGHAN) in children and adolescents with lip and mucosal freckling suggestive of PJS [48]. No new literature has been found to contradict their recommendation. There are no data reporting on the utility of genetic screening in the setting of a solitary PJ polyp, but testing of the *STK11* gene is a useful tool to clarify the risk of PJS, especially in children and younger adults.
**If the clinical diagnostic criteria for PJS are met, genetic germline screening of the STK11 gene is warranted regardless of age. A patient meeting the clinical criteria should be regarded as having PJS, even if an underlying causative germline variant is not identified.***Level of evidence: moderate**Strength of recommendation: strong*
**The detection rate of pathogenic STK11 variants in patients with a clinical diagnosis of PJS is high (up to 100%), using techniques that detect single nucleotide changes as well as larger deletions and duplications in the STK11 gene. Currently, there is no evidence for genetic heterogeneity in patients fulfilling the diagnostic clinical criteria for PJS without a germline STK11 PV. Thus pathogenic variants that cannot be identified by up-to-date methods in routine diagnostics should be considered in these cases.***Level of evidence: moderate**Strength of recommendation: strong*
**Based on recent recommendations of the European Society for Pediatric Gastroenterology Hepatology and Nutrition (ESPHGAN), genetic germline screening of the STK11 gene is warranted in children and adolescents with typical perioral pigmentation. Genetic screening may be considered in adults with isolated typical perioral pigmentation but is less likely to yield a pathogenic variant with increasing age.***Level of evidence: low**Strength of recommendation: moderate*
**Genetic germline screening of the STK11 gene is warranted in children and adolescents with one PJ polyp. Genetic screening may be considered in adults with a confident diagnosis of solitary polyp but is less likely to yield a pathogenic variant with increasing age.***Level of evidence: low**Strength of recommendation: moderate*
**If no pathogenic variant in STK11 can be identified in a patient not fulfilling the clinical diagnostic criteria for PJS, the patient should not be considered as having PJS.**                    *Level of evidence: low**Strength of recommendation: moderate*

### 4.2. Gastrointestinal Management

In 2019, the European Society of Gastrointestinal Endoscopy (ESGE) and European Society for Pediatric Gastroenterology Hepatology and Nutrition (ESPHAN) published guidelines including the luminal gastrointestinal management of PJS [48,49]. After reviewing these and confirming there is no more recent relevant literature regarding the gastrointestinal management of PJS, the EHTG guideline committee members endorsed these guidelines without modifications.
**Based on recent recommendations of the European Society of Gastrointestinal Endoscopy (ESGE), a baseline oesophagogastroduodenoscopy and colonoscopy is recommended at the age of 8 years in asymptomatic individuals with PJS. If polyps are detected at the baseline endoscopy, a 1–3 yearly interval based on phenotype for oesophagogastroduodenoscopy and/or colonoscopy is recommended. Routine oesophagogastroduodenoscopy and colonoscopy surveillance is recommended at the age of 18 if the baseline endoscopy is negative.***Level of evidence: low**Strength of recommendation: strong*
**Based on recent recommendations of the European Society of Gastrointestinal Endoscopy (ESGE), small bowel surveillance is recommended from the age of 8 years in asymptomatic individuals with PJS. A 1–3 yearly interval is recommended based on phenotype for small-bowel surveillance. Either MRI studies or video capsule enteroscopy is recommended for small-bowel surveillance.***Level of evidence: moderate**Strength of recommendation: strong*
**Based on recent recommendations of the European Society of Gastrointestinal Endoscopy (ESGE) elective polypectomy should be performed for small-bowel polyps > 15–20 mm to prevent intussusception. In a symptomatic patient, smaller polyps causing obstructive symptoms should be removed.***Level of evidence: low**Strength of recommendation: strong*
**Based on recent recommendations of the European Society of Gastrointestinal Endoscopy (ESGE), device-assisted enteroscopy for the removal of polyps is recommended. Based on phenotype, intraoperative enteroscopy could be considered.***Level of evidence: moderate**Strength of recommendation: strong*
**In case of symptoms, an oesophagogastroduodenoscopy, small bowel investigation, or colonoscopy should be performed earlier rather than waiting for routine surveillance.***Level of evidence: low**Strength of recommendation: strong*

Solitary PJ polyps are rare, but there are numerous case reports describing the identification of a solitary PJ polyp, which may occur at all sites of the gastrointestinal tract, with the exception of the esophagus. There are only two published case series of patients with solitary PJ polyps, which, although imperfect, represent the best data available. Oncel and colleagues described eight patients managed at the Cleveland Clinic [50]. Six of the eight patients were male, and the median age at diagnosis was 56 years. During a median follow up of 11.5 years (range 3–22), no patient developed a metachronous PJ polyp. One patient with a duodenal solitary PJ polyp developed a metachronous colorectal cancer and died 12 years after the diagnosis of the solitary PJ polyp. All patients in this series underwent pan-enteric imaging and had endoscopic surveillance. In addition, a full physical assessment was included. Therefore, a clinical diagnosis of PJS was robustly excluded. Genetic testing was not performed.

More recently, a larger case series from Japan has been reported [51]. This multicenter study reported on 51 patients (32/51 (63%) male) with a mean age of diagnosis of 66 years (range 32–92). The mean endoscopic follow up was 3 years (range 0.1–16 years). No patient developed a metachronous PJ polyp or GI cancer, although it was noted that 12 patients had had a GI tract cancer prior to the diagnosis of the solitary PJ polyp. There are some weaknesses in this study. It is not clear whether a systematic family history had been taken and whether complete physical assessment had been performed, being a retrospective multicenter study. Furthermore, 47/51 patients did not have small bowel evaluation, and 26/51 did not have any endoscopic surveillance.

Although the data above represent weak evidence, they suggest that there is no increased risk of metachronous PJ polyps or cancer, which in addition to the profiling data suggests that routine surveillance is not required once a diagnosis of solitary PJ polyp has been made. However, it is key that an appropriate assessment has been made to exclude PJS on clinical grounds; physical inspection for the typical mucocutaneous pigmented lesions, a full family history, and pan-enteric assessment (gastroscopy, small bowel imaging, and colonoscopy) are required before a diagnosis of solitary PJ polyp can be made. There are no data reporting on the utility of genetic testing in the setting of a solitary PJ polyp, but as stated above, genetic testing of the *STK11* gene is a useful tool to diminish the risk of PJS, especially in children and younger adults.
**For patients with a confident diagnosis of a solitary PJ polyp, routine endoscopic surveillance is not recommended.                     ***Level of evidence: low                     **Strength of recommendation: strong                     *

There are no data to address the question whether there is a role for haemoglobin testing in children with PJS. There is a wide literature of anemia (with or without overt GI bleeding) as a mode of presentation for a subsequent diagnosis of PJS, but there are no data regarding routine haemoglobin testing in children with PJS, let alone children under 8 years of age, which is the advised age to start gastrointestinal endoscopy. Therefore, we are unable to recommend that haemoglobin testing should routinely be performed as part of the surveillance of children with PJS. However, if there is any clinical suspicion regarding significant polyps (e.g., either due to history or a reduction in the centile on weight/height growth charts), checking the haemoglobin level may be a useful adjunct to guide us to the need for investigation with standard surveillance.
**Routine haemoglobin testing in children with PJS is not recommended, as there are no data reporting on its utility and outcome. Haemoglobin testing may be useful in the symptomatic setting.***Level of evidence: low**Strength of recommendation: weak*

### 4.3. Surgical Management

Intussusception occurs when a proximal segment of bowel and its mesentery slides into the lumen of the adjacent distal segment. In PJS, a polyp typically forms the hypomochlion that subsequently leads to intussusception due to bowel peristalsis. Intussusception is a surgical emergency leading to bowel ischemia, necrosis, and perforation when untreated (Figure 2).

In PJS, the risk of intussusception is estimated to be 44% by the age of 10 and about 50% by the age of 20% [52]. The risk of intussusception increases with increasing polyp size of 15 mm and larger [48]. The surgical reduction of intussusception should be undertaken without delay to avoid necrosis and resection of the small bowel. Usually, laparotomy is the safest option, but in selected, milder cases, laparoscopy can be considered. When ischemia is reversible, resection of the bowel should not be done but only a polypectomy. In addition, intraoperative enteroscopy through enterotomy is recommended to find and remove over 15 mm size polyps. If enteroscopy is not available, illumination and thorough palpation of the small bowel is recommended in order to palpate and remove larger polyps [53]. Up to 40% of PJS patients requiring laparotomy before the age of 18 will require a new laparotomy within 5 years of the first laparotomy [48]. The risk of malignant polyp during childhood is zero and low also during adulthood being 2.3–4.5% according to the literature [12,19,52,54].
**PJS patients with an episode of acute severe abdominal pain and/or suspicion of intussusception should urgently be referred to a surgical unit, preferably a dedicated center. If, after clinical and diagnostic evaluation the event of small bowel intussusception is not ruled out, emergency surgery (even in diagnostic intent) is recommended.***Level of evidence: moderate/low**Strength of recommendation: strong*
**At surgery, the preferred strategy of treating an intussusception is to dessuscept, if safe to do so. If successful, the polyp that acts as a hypomochlion should be removed by enterotomy with resection of the (pedunculated) polyp at the base. In addition, the entire small bowel should be critically inspected for further relevant polyps, and all polyps > 15 mm should be removed by enterotomy or by intraoperative enteroscopy. Depending on the distance between the polyps, an enterotomy in between polyps allowing for removal of multiple polyps via one enterotomy is preferred.***Level of evidence: moderate/low**Strength of recommendation: strong*

### 4.4. Pancreatic Management

In 2019, the International Cancer of the Pancreas (CAPS) Consortium formulated guidelines on pancreatic surveillance [55]. After reviewing the literature on pancreatic surveillance in PJS, the EHTG guideline committee members endorse the CAPS guidelines for this patient group.

Pancreatic cancer (PDAC) is the third most common tumor affecting PJS patients with a lifetime risk of 11–55% (Table 2) [12,13,16,17,18]. In a recent pancreas surveillance study in high-risk individuals (HRIs) by Abe et al. [56], the cumulative incidence of PDAC in the group with germline PV in known PDAC predisposing genes (including 12 PJS patients) was higher than in the familial risk (FPC) group. Bannon et al. also demonstrated that germline PVs in PDAC predisposing genes are highly prevalent in patients with early onset PDAC [57]. Nevertheless, in the previous version of this guideline, Beggs et al. did not recommend routine surveillance for pancreatic cancer in PJS because of a lack of sufficient evidence regarding its benefit and cost effectiveness; surveillance should be undertaken only in the framework of a clinical research study [5]. This concern and advice has also been voiced by others [58].

Few studies compared the diagnostic yield of EUS and MRI/CPRM in pancreatic surveillance. A high concordance of clinically relevant lesions’ detection between the two methods was described by Canto et al. [59]. Conversely, Harinck et al. demonstrated that, contrary to EUS, MRI was more sensitive for cystic lesions detection, with important limitations in solid lesions detection [60]. A meta-analysis performed by Signoretti et al. confirmed these results: the pooled prevalence of solid lesions detected by the EUS was higher compared with MRI (5.2% vs. 4.1%), while MRI demonstrated a higher yield for cystic lesions (22.4% vs. 16.6%), even if the pooled prevalence of surveillance target lesions was similar between EUS and MRI [61]. Therefore, these two methods might be considered complementary in pancreatic surveillance programs and tailored considering local expertise.
**Although PJS is considered a hereditary condition that carries some of the highest lifetime risks for developing pancreatic cancer, it should be discussed with patients that the benefits and harms of pancreatic cancer surveillance are not well established yet and under investigation. Therefore, it is recommended that surveillance is conducted at centers of expertise in the framework of a study or registry.***Level of evidence: moderate/low**Strength of recommendation: strong*
**Based on recent recommendations of the International Cancer of the Pancreas (CAPS) Consortium, patients with PJS are eligible for pancreatic surveillance in the framework of a study or registry, irrespective of patients’ family history of pancreatic cancer (PDAC), because of an estimated lifetime risk to develop PDAC of 11–55%.***Level of evidence: moderate/low**Strength of recommendation: strong*
**The recommendations for pancreatic surveillance of patients with PJS of the International CAPS Consortium are endorsed and should be followed.***Level of evidence: moderate/low**Strength of recommendation: weak*

In a multicenter prospective study, Konings et al. reported a very high incidence of cystic lesions both in individuals with FPC (61%) and PV carriers (47%), including 11 PJS patients [62]. They also demonstrated that while individuals with FPC were significantly more likely to have pancreatic cysts 10 mm or greater than PV carriers, the cysts in the latter group were more likely to progress during follow-up (PDAC incidence 2%). Subsequently, Barnes et al. performed pancreatic screening with 3.0 T MRI routinely in a group of 65 HRIs (including one PJS patient) and reported pancreatic abnormalities in 28 (43%), which were all cystic lesions [63]. There was no association with age, genetic disposition, or estimated PancPRO PC risk. In 354 HRIs (including 10 PJS patients) enrolled prospectively in CAPS studies from 1998 to 2014, 14 HRI (4%) with solid hypoechoic masses > 1 cm or nodules < 1 cm at baseline and 4 (1.1%) with both cysts and solid lesions were found [64]. The remaining 151 (43%) HRIs had one or more cystic lesions at baseline and 49 (14%) had three or more cysts. The mean size of the largest cyst at baseline was 8 mm (range 1.6–28 mm). The overall detection rate for PDAC or a high-grade dysplasia in 354 HRIs during the 16-year follow-up was 7%, including prevalent and incident neoplasms. HRIs with neoplastic progression were more likely to have multiple cysts (three or more) at baseline compared to non-progressors (PDAC 36% and high-grade precursor lesions 80%, versus others 11%, *p* < 0.0001), even after adjusting for other factors (HR 4.85, 95% CI 2.02–11.64). In particular, the presence of a solid mass, mural nodule, thickened cyst wall, rapid cyst growth rate, and an MPD (main pancreatic duct) dilated to >5 mm at any time during surveillance were associated with the development of PDAC or high-grade precursor neoplasm, both at univariate and multivariate analysis.
**Prevailing regional pancreatic cyst surveillance guidelines should be carried out for cyst follow-up and management in PJS patients.***Level of evidence: moderate/low**Strength of recommendation: weak*
**Any significant abnormal finding during surveillance should be discussed in a multidisciplinary panel.***Level of evidence: low**Strength of recommendation: strong*

Two surgical approaches have been proposed for HRIs with pathologic findings identified during surveillance: the radical approach (total pancreatectomy) and the conservative (partial resection) surgical therapy. The main advantage of total pancreatectomy is radical removal of all pancreatic high-risk parenchyma, given the multifocality of precancerous pancreatic lesions in HRIs [65,66]. However, it has a significant morbidity due to exocrine and endocrine pancreatic insufficiency. Pancreatic islet transplantation has been used to solve that problem, but it is associated with the potential risk of neoplastic cell seeding [67,68]. Partial pancreatectomy depends on the localization of the pancreatic lesion [69]. It has the risk of PDAC development in pancreatic remnant: indeed, HRIs develop multiple precursors throughout their pancreas, and those who undergo partial pancreatic resection for IPMN can have concomitant high-grade PanIN, sometimes making secondary total pancreatectomy necessary [70,71]. There is no evidence to support the more radical approach unless there are concerning lesions affecting multiple regions of the gland. There was also no CAPS consensus that surgical resection was indicated for less worrying lesions, such as suspected IPMN of 2 cm or with mild main pancreatic duct dilatation [55].

In a meta-analysis of 16 studies by Paiella et al. including a total of 1551 FPC patients (syndromic HRIs were excluded), 30 subjects (1.82%) received a diagnosis of PDAC, PanIN3, or HGD-IPMNs [72]. Therefore, the pooled proportion of screening goal achievement (SGA) was high and equal to 1.4% (95% CI 0.8–2, *p* < 0.001, I2 = 0%), while the pooled proportion of overall surgery was 6% (95% CI 4.1–7.9, *p* < 0.001, I2 = 60.91%), and that of unnecessary surgery was 68.1% (95% CI 59.5–76.7, *p* < 0.001, I2 = 4.05%). These results suggest that the probability of proceeding to surgery during surveillance is non-negligible, and unnecessary surgery is a potential negative outcome. Another meta-analysis was performed on 13 studies, including 90 HRIs (PJS patients in seven out of 13 studies) by de Mestier et al. and demonstrated that the surgical resection specimen revealed a pre/malignant lesion in 38 HRIs (42.2%), including 20 PDAC (22.2%) [73].

A recent multicenter international study was conducted through the CAPS Consortium Registry to examine the diagnostic yield and outcomes of HRIs who underwent surgical resection or progressed to invasive cancer under surveillance and the characteristics of patients who developed high-risk neoplastic precursor lesions or PDAC [74]. Of 76 high-risk individuals identified in 11 surveillance programs, 71 had undergone surgery (three PJS patients) and five had been diagnosed with inoperable PDAC (one PJS patient). EUS detected most lesions (87%). A total of 93 suspicious lesions were detected by EUS in the 71 patients who underwent resection, 44 (47%) were cystic, 33 (35%) were solid, and 16 had another appearance. Distal pancreatectomy was performed in 36 patients (51%), and there were no surgery-related deaths. At surgery, 32 (45%) patients had PDAC or a high-risk precursor (19 PDAC, 4 main-duct IPMN, 4 branch-duct IPMN, 5 PanIN-3); however, only three of the 19 PDACs had T1 status. The other 39 patients (55%) had lesions thought to be associated with a lower risk of neoplastic progression. Age at least 65 years, female sex, carriage of a gene mutation, and location of a lesion in the head/uncinate region were associated with high-risk precursor lesions or PDAC lesion. Of the 71 high-risk individuals who underwent surgery, 59 (83%) were still alive after a mean follow-up of 54, 3 months, and of the 12 patients who died, eight deaths were PDAC-related. The survival of high-risk patients with no or low-risk lesions did not differ significantly from that of patients with high-risk neoplastic precursor lesions.

Another recent study carried out by Canto et al. evaluated HRIs (total number of PJS patients was not reported) outcomes after pancreatic resection during surveillance: 354 asymptomatic HRIs enrolled prospectively in CAPS studies from 1998 to 2014 and underwent surveillance for at least 6 months [75]. The authors demonstrated that 48 HRIs (13.6%) had 57 operations for suspected pancreatic lesions: 48 were initial (16 Whipple’s procedures, 26 distal pancreatectomy, 6 total pancreatectomy) and 9 s surgery procedures (5 distal pancreatectomy, 4 Whipple’s procedures) for a new lesion after a median of 3.8 years (IQR 2.5–7.6). Eleven PDAC (two stage I and eight stage II cancers) and 10 high-grade precursor lesions (6% of the 354 cohort) were diagnosed and surgically treated during the 19-year study period. The one-year overall survival was 90%, while 5-year overall survival was 60% for PDAC patients. The median length of hospital stay for the 48 HRIs with initial surgery procedures was 7 days (IQR 5–11), although patients who have had total pancreatectomy required a median of 11.5 days (IQR 8.5–13.3). Overall, postoperative complications developed in 17 (35.4%), with zero 90-day mortality. Patients receiving Whipple’s procedure as initial surgery had more complications (62.5%) compared to the other two groups (*p* = 0.02), in particular delayed gastric emptying (37.5%, *p* = 0.01). Postoperative diabetes developed in 20% HRIs who underwent partial pancreatectomy with no difference between distal and Whipple surgery, while it developed in 100% of HRIs receiving total pancreatectomy. No intra-abdominal hemorrhage was observed.

When screening is negative in HRIs, prophylactic pancreatectomy is not indicated in view of its significant morbidity and the potential mortality even in experienced hands, mainly with pancreatoduodenectomy [65,76].
**According to the recent recommendations of the International CAPS Consortium, a (partial) pancreatectomy should be performed in case of detection of: (i) a solid lesion ≥ 10 mm (except biopsy-proven or highly suspicious to be neuroendocrine, autoimmune, or other benign conditions); (ii) IPMN in case of a mural nodule, an enhanced solid component, symptoms (including pancreatitis, jaundice, pain), thickened/enhanced cyst walls, abrupt change in pancreatic duct with distal pancreatic atrophy, or a main pancreatic duct ≥ 10 mm.***Level of evidence: moderate/low**Strength of recommendation: strong*
**Due to its significant morbidity and potential mortality even in experienced hands, a total pancreatectomy is not recommended for a localized lesion.***Level of evidence: low**Strength of recommendation: strong*
**Prophylactic pancreatectomy is not recommended because of the significant associated morbidity and potential mortality, even in experienced hands.***Level of evidence: low**Strength of recommendation: strong*

### 4.5. Breast Management

Estimates for the lifetime risk of breast cancer (BC) in women with PJS vary widely from 19.3% to 54%, which is probably due to the small sample sizes in most studies (Table 2) [12,13,14,17,18]. Over the last decade, only seven cohort studies on BC risk in PJS were published, including more than ten women (see Appendix A). In 2011, data from a Dutch, partly prospective cohort study on 133 PJS patients (54 families; 69 females) reported six cases of BC and a BC age with a range of 46–61 years [15]. An Italian retrospective cohort study on PJS patients reported two cases of BC (at the age of 48 and 52 years) among 61 female PJS patients [17]. Data from China in 2017 showed an RR of 28 (CI 7–113) in a cohort study of 336 PJS patients (155 females) [19]. In 2018, Chiang et al. and Fostira et al. reported respectively BC in 2/8 and 3/10 female PJS patients [39,77]. Lipsa et al. reported BC in 4/7 women with a PV in *STK11* and in 5/8 women suspected of having PJS (based on mucocutaneous pigmentation) [78]. Multiple studies indicated cases with bilateral BC, and one case of a male PJS patient developing BC was described [12,13,77,78,79]. A few studies stratified the risk of breast cancer in female PJS patients; BC risk was 5–12.7% at age 40, 11–24% at age 50, and up to 24–54% at age 60–70 years [12,13,14,17,25,80]. In the meta-analysis and three systematic reviews, the mean age at BC diagnosis ranged from 37 to 45 years [12,13,14,17]. In most studies, the mean age at breast cancer diagnosis was >30 years of age. However, breast cancer has been reported in PJS patients in their early 30s and in some even <30 years of age [12,18,78].

No clinical trials on breast surveillance protocols for women with PJS have been published. Although PJS is mentioned in guidelines on breast cancer surveillance for individuals at high risk of developing cancer, there are limited recommendations for PJS patients specifically. MRI starting at the age of 25–30 years is most often recommended, and several authors refer to the NCCN guidelines for hereditary breast and ovarian cancer, which recommend mammogram and breast MRI annually and clinical breast evaluation every six months, all starting at age 25 [5,81,82]. Several authors remarked that the highest estimates of BC risk in women with PJS overlap with BC risk in BRCA mutation carriers, suggesting the same surveillance strategy for those high-risk patients [83,84]. Boetes et al. emphasized the role of screening with breast MRI in asymptomatic females at high risk of developing BC, since MRI has a higher sensitivity of more than 70% compared with the sensitivity up to 40% of mammography alone [85]. Sensitivity of mammography is especially lower with dense breasts, which are more common in younger women. It seems reasonable to start screening by MRI at 25 years of age, but starting at a younger age warrants consideration based on family history. Breast self-examination, although not proven effective for the detection of early cancer, can also raise breast awareness from a younger age. No data are available on prophylactic mastectomy in PJS patients.
**The following breast surveillance is recommended in female PJS patients: Raising awareness at age 18 years e.g., by starting breast self-examination; Clinical breast exam every 6–12 months starting at age of 25 years; Annual breast contrast MRI screening (or breast ultrasound if MRI contraindication or unavailability) at age 25–30 years; Annual mammogram with consideration of tomosynthesis and ultrasound for dense breast and annual breast contrast MRI at age 30–50 years; Annual mammogram with consideration of annual breast contrast MRI for dense breast pattern at age 50–75 years; Management should be considered on an individual basis from age > 75 years.***Level of evidence: low**Strength of recommendation: moderate*
**The optimal breast surveillance strategy in female PJS patients remains debated and the benefits of surveillance remain to be established. Therefore, it is recommended that surveillance is conducted at centers of expertise in the framework of a study or registry.***Level of evidence: low**Strength of recommendation: strong*
**As evidence for its benefit is lacking, prophylactic mastectomy is currently not recommended for female PJS patients. Risk reducing mastectomy should be discussed in a multidisciplinary setting also taking into account family history and other clinical factors.***Level of evidence: low**Strength of recommendation: moderate*

### 4.6. Gynecological Management

The risk of gynecological cancer is increased in women with PJS with current estimations ranging from 18% to 50% by the age of 50 years (Table 2) [12,13,16,18]. In the meta-analysis by Giardello et al., based on 6 publications and 107 females from 72 PJS families, 2 uterine cancers, 4 ovarian cancers, and 3 cervical cancers were detected. The risks for uterine and ovarian, but not for cervical cancer, were significantly increased [12]. In the cohort study by Hearle et al., 226 female PJS patients from European centers were included [13]. Nine women developed a gynecological cancer: two ovarian, two uterine, and five cervical cancers. The risk for gynecological cancer was not significantly increased by the age of 40, but by the age of 50, the risk for gynecological cancer was 8-fold (CI 4–199), and by 60, it was 18-fold (CI 9–34). Mehenni et al. collected data about 149 patients with PJS and *LKB1* germline mutations from four different cancer institutions [14]. Seven out of the 73 women developed gynecological cancer: four uterine cancers between 35 and 45 years of age, and three ovarian cancers between 22 and 38 years of age. The type of the gynecological cancers was not described in these papers, limiting further analysis. Resta et al. gathered 61 female *STK11* germline PV carriers from 16 institutes. Seven gynecological cancers were detected [17]. Of the four cervical cancers, three were mucinous adenocarcinomas. Of the ovarian cancers, one was a malignant SCTAT (sex cord tumor with annular tubules) at the age of 37 years, one was a borderline mucinous ovarian cancer at the age of 18 years, and one ovarian cancer at the age of 41 was not specified. Van Lier et al. published a systematic review on cancer risks in PJS patients including published reports until February 2009 [81]. Their review included 20 cohort studies, including the above-mentioned papers by Hearle et al. and Mehenni et al., and the meta-analysis by Giardello et al., which was already presented here. Based on four cohort studies, the cumulative risk of any gynecological cancer by the age of 50 was between 10 and 20%. This paper also gives expert opinion-based guidelines for gynecological cancer surveillance, suggesting annual pelvic examination, Pap smear, transvaginal ultrasound, and CA-125 measurement starting from the age of 25–30 years. Ishida et al. reported cancer occurrence in a total of 313 female Japanese PJS patients in their meta-analysis [18]. Fifty-four women were reported to have a uterine carcinoma of which 52 (96%) were cervical adenocarcinomas. Of these 52 cervical adenocarcinomas, 30 were “minimal deviation adenocarcinoma”, which is a rare variant of cervical mucinous adenocarcinoma that is also known as adenoma malignum. The risk of developing any gynecological cancer was 14.6% at 30 years, 29.2% at 40 years, 49% at 50 years, and 55.4% at 60 years.

There are no data on prospective surveillance programs with a systematic approach to gynecological cancer surveillance. Van Lier et al. published results from their program, which included all PJS patients from two Dutch hospitals [15]. The patients were prospectively followed from 1995 to July 2009. The cohort included 69 females. During the surveillance period, six gynecological cancers were detected: two malignant Sertoli cell ovarian tumors at the age of 16 and 37 years, one small cell ovarian carcinoma at the age of 30 years, two cervical minimal deviation adenocarcinomas at the age of 35 and 72 years, and one cervical cancer not specified at the age of 45 years. The paper does not describe how the patients were followed and how the tumors were detected. The existing literature gives no evidence-based data for recommendation of surveillance.

In conclusion, cervical adenocarcinoma, in particular minimal deviation adenocarcinoma (adenoma malignum), is the most frequently reported gynecological cancer in women with PJS. The risk for ovarian cancer is also increased, but the histology of the ovarian cancers is not well reported. Based on the literature, the ovarian cancer risk seems to apply to non-epithelial ovarian cancer (SCTAT), with the risk of the more common epithelial ovarian cancer not increased. The reports do not suggest that the risk of endometrial cancer is increased in women with PJS. In the absence of evidence-based data on gynecological surveillance, our recommendation is based on expert opinion and current knowledge of gynecological cancer risks in women with PJS. The detection of minimal deviation adenocarcinoma from Pap smears or from clinical features is difficult and requires a high index of suspicion. These tumors are not caused by high-risk human papillomavirus (HPV), and therefore, routine cervical screening triaged by the presence of HPV may fail to detect them. Except for vaginal ultrasound examination of the ovaries, there are no good screening tests for non-epithelial ovarian tumors, which constitute the major ovarian cancer risk. There are no data on the usefulness of tumor markers (e.g., CA125) for ovarian cancer surveillance in PJS patients.
**Expert gynecological surveillance should be offered to female patients with PJS, irrespective of their family history of gynecological cancer, because of an estimated lifetime risk of specific gynecological tumors of 18–50%.***Level of evidence: low**Strength of recommendation: moderate*
**It is recommended that female PJS patients are counseled regarding specific gynecological cancer risks, red flag symptoms, contraceptive choices, and family planning by a PJS specialist at 18–20 years of age.***Level of evidence: low**Strength of recommendation: moderate*
**It is recommended that female PJS patients have annual gynecological examinations from the age of 25 years. In addition to cervical screening as performed in population-based screening programs that run in many countries, gynecological surveillance in female PJS patients should be focused on the detection of cervical adenocarcinomas, in particularly minimal deviation adenocarcinoma, and rare non-epithelial ovarian tumors. Surveillance for cervical adenocarcinomas should involve speculum examination and cervical screening ("Pap smear") including cytology even in an HPV-negative sample. Surveillance for non-epithelial ovarian cancers should involve bimanual pelvic examination with a transvaginal ultrasound in case of suspicion of a pelvic mass. CA125 testing is not indicated.***Level of evidence: low**Strength of recommendation: moderate*
**The optimal gynecological surveillance strategy in female PJS patients remains debated and the benefits of surveillance remain to be established. Therefore, it is recommended that surveillance is conducted by a gynecologist who is experienced in the particular cancer risks that PJS patients face in the framework of a study or registry.***Level of evidence: low**Strength of recommendation: strong*

There is no literature on experience with prenatal genetic diagnosis (PND) or preimplantation genetic diagnosis (PGD) in PJS. Wang et al. describe a positive prenatal genetic test for PJS with continuation of the pregnancy; PGD for a subsequent pregnancy of the PJS patient was suggested [86]. Woo et al. performed a questionnaire survey about psychological wellbeing among 38 PJS patients and their relatives: 40% altered reproductive choices because of PJS and 33% were reluctant to have children due to the risk of PJS [87]. They emphasis the need for counseling on reproductive options for PJS patients. Van Lier et al. performed a questionnaire survey among 52 PJS patients on family planning: in 29%, PJS influenced decisions about family planning, 19% did not want children because of PJS, termination of pregnancy was considered acceptable by 15% and PGD was considered acceptable by 52% [88]. Based on this and the experience with other cancer predisposition syndromes, PJS should be considered an indication for PND and PGD, and PJS patients should be counseled about their reproductive choices.


**Peutz-Jeghers syndrome can be an indication for Prenatal Genetic Diagnosis (PND) and Preimplantation Genetic Diagnosis (PGD) and these options should be discussed with PJS patients in whom a STK11 pathogenic variant has been identified.**

*Level of evidence: low*

*Strength of recommendation: strong*


## 5. Conclusions

The evidence base for recommendations regarding the management of PJS remains poor, and collaborative studies are required to provide better data on this rare condition. Within these restrictions, multisystem, clinical management recommendations for PJS have been reviewed and updated. EHTG supports the concept of continuous revision of these recommendations as a concept of “dynamic” guidance. In the event of relevant literature providing evidence for a better management strategy, the corresponding recommendation will be revised accordingly. In a multidisciplinary management program, all parties (including patients) are welcome to approach EHTG and request a revision.

## Figures and Tables

**Figure 1 jcm-10-00473-f001:**
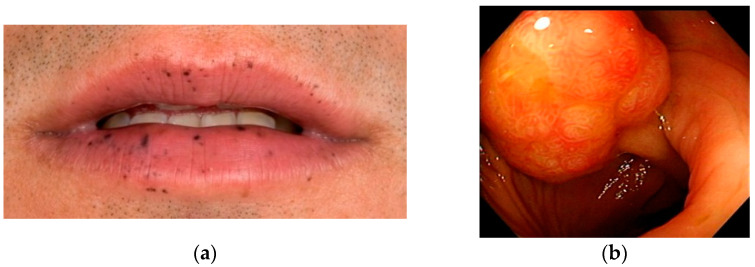
Pigmentations (**a**) [6] and polyp (**b**) characteristic for Peutz–Jeghers syndrome.

**Figure 2 jcm-10-00473-f002:**
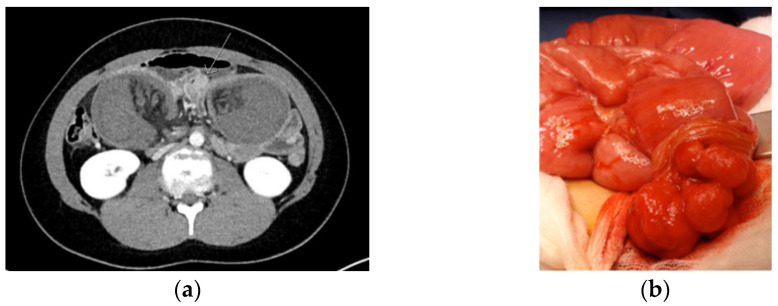
CT image of intussusception (**a**) and to be resected polyp (**b**) in a Peutz–Jeghers syndrome patient.

**Table 1 jcm-10-00473-t001:** Diagnostic clinical criteria for Peutz–Jeghers syndrome (PJS).

**Tomlinson and Houlston 1997** [3]
1: Two or more PJS polyps in the gastrointestinal tract or
2: One PJS polyp in the gastrointestinal tract, together with either classical PJS pigmentation or a family history of PJS
**WHO Criteria 2000** [4]
A: A positive family history of PJS and
1: Any number of histologically confirmed PJS polyps or
2: Characteristic prominent mucocutaneous pigmentation
B: A negative family history of PJS and
1: Three histologically confirmed PJS polyps or
2: Any number of histologically confirmed PJS polyps and characteristic prominent mucocutaneous pigmentation
**Beggs et al. 2010** [5]
1: Two or more histologically confirmed PJS polyps or
2: Any number of PJS polyps in an individual who has a family history of PJS in close relative(s) or
3: Characteristic mucocutaneous pigmentation in an individual who has a family history of PJS in close relative(s) or
4: Any number of PJS polyps in an individual who also has characteristic mucocutaneous pigmentation

**Table 2 jcm-10-00473-t002:** Studies on cumulative cancer risks in Peutz–Jeghers syndrome.

Study	N	gac	smbc	crc	Gastroint. Cancer	pac	bc	utc	ovc	cx	Gynecol. Cancer	All	At Age (Years)
Gardielo et al., 2000 [12], meta-analysis	210	29	13	39		36	54	9	21	10		93	64
Hearle et al., 2006 [13], cohort study	419				57	11	45				18	85	70
Mehenni et al., 2006 [14], cohort study	149				63 *	18 #						67	70
Van Lier et al., 2011 [15], cohort study	133				51 **							76	70
Korsse et al., 2013 [16], cohort study	144					26							70
Resta et al., 2013 [17], cohort study	119			12		55	24			23		89	60–65
Ishida et al., 2016 [18], meta-analysis	583	24	10–14	36		29	19	47 ***	10			83	70
Chen et al., 2017 [19], cohort study	336			28								55	60

gac = gastric cancer; smbc = small bowel cancer; crc = colorectal cancer; gastroint. = gastrointestinal; pac = pancreas cancer; bc = breast cancer; utc = uterus cancer; ovc = ovary cancer; cx = cervix cancer; gynecol. = gynecological; * including biliary tract cancer; ** including pancreas and biliary tract cancer; *** including adenocarcinoma cervix; # at age 50 years.

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
