# Peer review of "The Management of Peutz–Jeghers Syndrome: European Hereditary Tumour Group (EHTG) Guidelineâ€"

_jcm, 2021, doi:10.3390/jcm10030473_

Round 1
Reviewer 1 Report
The European Hereditary Tumor Group made a guideline for Peutz-Jeghers syndrome (PJS) to manage the clinically important situation about the PJS. The manuscript is written in a sophisticated manner by European expert doctors. I have raised only a few comments.
Minor
- About Table2, although pac is included in gastrointestinal cancer, the word ‘gastrointestinal’ usually does not include the pancreas. Pancreas cancer is categorized as cancers in the pancreaticobiliary system.
- If the images of a typical presentation about PJS are available, it is useful for physicians who read this guideline. Not all doctors are familiar with For example, pictures of hyperpigmentation, the gross picture of PJS polyp, and CT findings of intussusception, etc.
- Although the statements are written in a bold and italic manner, it is still a little difficult to read. If possible, the layout should be modified to highlight the main statements. Moreover, if a table of contents is available, it is useful for readers because physicians in the various field will read this guideline.
- About 3.1 Clinical genetic management, if necessary, it might be a good idea to write about genetic counseling before genetic testing. Further, I think the way of genetic testing for children should be written (informed assent).
- About PJS pigmentation, the efficacy of laser treatment has been reported which is useful for patients who are suffering from appearance caused by PJS. (Li Y, Tong X, Yang J, et al.: Q-switched alexandrite laser treatment of facial and labial lentigines associated with Peutz-Jeghers syndrome. Photodermatol Photoimmunol Photomed 2012; 28: 196–199.
- About 3.2 Gastrointestinal management, dose MRI studies mean MR enterography or simple MRI?
Author Response
Reviewer 1:
The European Hereditary Tumor Group made a guideline for Peutz-Jeghers syndrome (PJS) to manage the clinically important situation about the PJS. The manuscript is written in a sophisticated manner by European expert doctors.
We thank the reviewer for these positive remarks.
I have raised only a few comments.
Minor
1. About Table2, although pac is included in gastrointestinal cancer, the word ‘gastrointestinal’ usually does not include the pancreas. Pancreas cancer is categorized as cancers in the pancreaticobiliary system.
Reply: This is adjusted in Table 2
2. If the images of a typical presentation about PJS are available, it is useful for physicians who read this guideline. Not all doctors are familiar with For example, pictures of hyperpigmentation, the gross picture of PJS polyp, and CT findings of intussusception, etc
Reply: Pictures of the characteristic pigmentations, PJS polyps and a CT image on intussusception are added.
3. Although the statements are written in a bold and italic manner, it is still a little difficult to read. If possible, the layout should be modified to highlight the main statements. Moreover, if a table of contents is available, it is useful for readers because physicians in the various field will read this guideline.
Reply: The lay-out of the recommendations and statements has been clarified by introducing the following sections, by adapting the numbering added by the journal and by putting the statement in boxes as suggested by reviewer 3.
4. About 3.1 Clinical genetic management, if necessary, it might be a good idea to write about genetic counseling before genetic testing. Further, I think the way of genetic testing for children should be written (informed assent).
Reply: We agree that all genetic testing should be done with informed consent or informed assent in the case of children. Since genetic counseling was not addressed by the current literature review, it was not included in the clinical genetic management section. In the introduction genetic counseling is added as an integral part of genetic testing. Since this is a “dynamic guideline“ more attention will be given to genetic counseling in a following literature revision.
5. About PJS pigmentation, the efficacy of laser treatment has been reported which is useful for patients who are suffering from appearance caused by PJS. (Li Y, Tong X, Yang J, et al.: Q-switched alexandrite laser treatment of facial and labial lentigines associated with Peutz-Jeghers syndrome. Photodermatol Photoimmunol Photomed 2012; 28: 196–199
Reply: The management of pigmentations was not addressed in the current literature review, but can also be included in the following edition of this dynamic guideline.
6. About 3.2 Gastrointestinal management, dose MRI studies mean MR enterography or simple MRI?
Reply: In the ESGE guidelines it was left as MRI studies to include both MR enterography and MR enteroclysis. There are data to support both but no comparisons between the two. Some feel that enteroclysis adds little to enterography but in the absence of a head to head study, it was felt that leaving either option open for each centre to decide on their preference was a pragmatic approach, given the evidence base upon which the stamen was made. In this guideline we have agreed to support that ESGE guideline, as there were no new data that needed to be considered.
Reviewer 2 Report
As the author points out, there is currently little scientific data on this disease. We believe that this guideline will be a useful indicator for the treatment of this disease.
Author Response
Reviewer 2:
As the author points out, there is currently little scientific data on this disease. We believe that this guideline will be a useful indicator for the treatment of this disease.
We thank this reviewer for the kind remark.
Reviewer 3 Report
- “http://www.ehtg.com/guidelines” does not exist: did you mean https://ehtg.org/guidelines/ ?
- Where is “*” note in Table 2?
- Explain better the meaning of “cumulative cancer risks” in Table 2: eosc 0,5 means these patients are protected by eosc? Can you explain the risk as a relative risk respect to general population?
- Why “Cancer risks in Peutz-Jeghers syndrome” is under Methods section?
- Why “Cancer risks in Peutz-Jeghers syndrome” is without a number and “Clinical genetic management” is preceded by “3.1” ?
- What about the management of Sertoli tumors of the testes? Airway polyps?
- Include the statements in a box
Author Response
Reviewer 3:
- http://www.ehtg.com/guidelines” does not exist: did you mean https://ehtg.org/guidelines/ ?
Reply: The reviewer is right and this was corrected.
- Where is “*” note in Table 2
Reply: Table 2 was adjusted.
- Explain better the meaning of “cumulative cancer risks” in Table 2: eosc 0,5 means these patients are protected by eosc? Can you explain the risk as a relative risk respect to general population?
Reply: Since cumulative cancer risks are more suitable for clinical practice, we chose to give an overview of these. In the text we now added that the overview is on cumulative cancer risks. Since oesophagus cancer risk is only described once by Gardiello et al. not very increased and confusing, this was removed from the table.
- Why “Cancer risks in Peutz-Jeghers syndrome” is under Methods section?
Reply: In fact we did not place “Cancer risks in Peutz-Jeghers syndrome” under the Method section, but as a separate paragraph. However, this was unclear due to a lay-out change by the journal. We changed the lay-out to make this clear.
- Why “Cancer risks in Peutz-Jeghers syndrome” is without a number and “Clinical genetic management” is preceded by “3.1” ?
Reply: This number was added by the journal. We suggested another numbering.
- What about the management of Sertoli tumors of the testes? Airway polyps?
Reply: We agree with the reviewer that many interesting issues regarding PJS deserve attention. Since this is a “dynamic guideline” the mentioned issues may be addressed in a following revision. Please note however that this has been addressed in a recent international paediatric guideline on PJS form ESPHGAN (JPGN 2019;68: 442–452).
- Include the statements in a box
The statements have been put in boxes.
Round 2
Reviewer 3 Report
Thank you very much for the corrections